# Preventive Behaviors and Information Sources during COVID-19 Pandemic: A Cross-Sectional Study in Japan

**DOI:** 10.3390/ijerph192114511

**Published:** 2022-11-04

**Authors:** Manae Uchibori, Cyrus Ghaznavi, Michio Murakami, Akifumi Eguchi, Hiroyuki Kunishima, Satoshi Kaneko, Keiko Maruyama-Sakurai, Hiroaki Miyata, Shuhei Nomura

**Affiliations:** 1Department of Health Policy and Management, School of Medicine, Keio University, 35 Shinanomachi, Shinjyuku-ku, Tokyo 160-8582, Japan; 2The Tokyo Foundation for Policy Research, 3-2-1 Roppongi, Tokyo 106-6234, Japan; 3Medical Education Program, Washington University School of Medicine, 660 S Euclid Ave, Saint Louis, MO 63110, USA; 4Division of Scientific Information and Public Policy, Center for Infectious Disease Education and Research, Osaka University, 2-8 Yamadaoka, Suita City 565-0871, Japan; 5Department of Global Health Policy, Graduate School of Medicine, The University of Tokyo, 7-3-1 Hongo, Bunkyo-ku, Tokyo 113-8654, Japan; 6Center for Preventive Medical Sciences, Department of Sustainable Health Science, Chiba University, 1-33 Yayoi-cho, Inage-ku, Chiba 263-8522, Japan; 7Department of Infectious Diseases, St. Marianna University School of Medicine, 2-16-1 Sugao, Miyamae-ku, Kawasaki 216-8511, Japan; 8Department of Ecoepidemiology, Institute of Tropical Medicine, Nagasaki University, 1-12-4 Sakamoto, Nagasaki 852-8523, Japan

**Keywords:** preventive behaviors, COVID-19, information sources, social media

## Abstract

Background: individual preventive behaviors are one of the key measures needed to prevent the spread of COVID-19. This study sought to identify the factors associated with the adoption of COVID-19 preventive measures, focusing specifically on information sources. Methods: we conducted a nationally representative cross-sectional survey of 30,053 Japanese adults in February 2021. The survey asked about socioeconomic, health-related, and psychological characteristics, attitudes toward immunization, and the use of information sources regarding COVID-19. We have constructed multivariable logistic regression to estimate the factors associated with the adoption of three preventive measures: 3Cs avoidance, hand hygiene and respiratory hygiene. Results: socioeconomic variables, psychological variables, and the use of information sources are significantly associated with the adoption of preventive measures. The more information sources one uses, the more likely one is to adopt preventive measures. Trust in healthcare professionals is positively associated with adopting preventive measures. On the other hand, negative correlations between trust in social media and preventive behaviors were observed. Conclusions: encouraging access to multiple information sources, utilizing communication channels, and modifying messaging according to target groups are essential to promote COVID-19 preventive measures.

## 1. Introduction

As of June 2022, more than 530 million cases of COVID-19 and 6.3 million deaths have been confirmed worldwide [1]. Although many vaccination campaigns against COVID-19 are ongoing, vaccination coverage is slow to increase [2]. Furthermore, new variants of the virus continue to emerge around the world. Recent studies have indicated that although the vaccine has maintained its effectiveness in preventing severe disease, the effectiveness against infections with new variants such as Omicron may decrease over time [3,4]. In addition, the Japanese legal system does not allow forced lockdowns or penalties for violations [5,6]. Systematic reviews have shown that preventive measures such as hand and respiratory hygiene are effective in reducing infection rates [7]. Hence, individual preventive behavior is a crucial way to slow the spread of the virus [8].

According to previous research, the decision whether or not to adopt preventive measures against infectious diseases is influenced by numerous factors [9,10]. Firstly, socioeconomic variables such as age, gender, and educational background have a great impact on the adherence to the preventive measures [11]. Second, psychological variables influence the action both negatively and positively [12,13,14]. In particular, the degree of anxiety is strongly correlated with the adoption of preventive measures as well as with the source of COVID-19 information and its trust level [15,16,17]. Knowledge and understanding of the disease are also correlated with preventive behaviors [18,19,20,21]. Trust is one of the key predictors of preventive behaviors in health area [22]. According to Nakayachi and Cvetkovich, trust is composed of competence, fairness, and salient value similarity [23]. Trust influences the perception of risk and benefit, and, accordingly, it has a powerful impact on decision making [24,25]. Thus, trust mediates the relationship between information use and health behaviors [26,27,28]. A previous study found a positive association between trust in formal information sources and preventative behaviors [29]. Previous research in Australia also identified that trust in health care professionals and scientists was linked to greater engagement in protective behaviors [30]. Hence, understanding the dynamics of how trust in information sources influences preventive behaviors is essential for developing a COVID-19 health messaging strategy. Additionally, a detailed analysis of social media and the disease is ongoing in general, but further research is needed to examine the effects [31,32]. Finally, prior studies have shown that health literacy has a positive influence on the adoption of preventive measures [33,34]. However, findings differ depending on the region [35,36]. The most recent evidence was generated in the United States and Europe; thus, insights from other regions remain limited. Many studies also point to the issues of representativeness due to the lack of large sample sizes, and the determinants of the preventive behavior analyzed vary from study to study.

In Japan, there are some previous studies that have analyzed the relationship between COVID-19 preventive behaviors and specific variables [5,6,37,38,39,40,41,42]. However, not many studies have comprehensively examined the relationship between preventive measures and various factors. In addition, most studies were conducted during the early stages of the pandemic. Our study was conducted during the second declaration of a state of emergency before COVID-19 vaccination of the public began in Japan, when preventive measures were crucial to slowing the spread of COVID-19. The present study comprehensively investigated various factors in a large sample size of 30,000 people.

In addition, most previous research has focused on the types of information sources trusted, and little is known about the number of information sources used [6]. Prior research showed that those with low levels of understanding and concern about COVID-19 use few information sources, and they tend not to take protective action [43]. If so, it can be hypothesized that the diversity of the information sources might affect the adherence to preventive measures. In order to investigate this, the present study will evaluate the relationship between the number of information sources used and COVID-19 prevention practices.

In brief, the objectives of this study are as follows: first, to comprehensively examine the relationship between various variables including socioeconomic and psychological variables and COVID-19 preventive behaviors; second, to identify how the use and trust of information sources are linked to the adoption of preventive measures.

## 2. Materials and Methods

### 2.1. Study Population

This study uses the same dataset as Nomura et al., Yoneoka et al., and Adachi et al., and further details are described elsewhere [17,44,45]. Survey respondents were collected through an online survey company, Cross Marketing, Inc. [46]. As of 2021, the company had approximately 4.7 million panel members with diverse demographic, geographic and socioeconomic characteristics. Panel registration is voluntary. Those who respond to surveys are awarded “points” based on the volume of responses they provide, and the points can be used to purchase products and services from partner companies. It is thought to be effective in reducing the effect of the selection bias because it may motivate those who are not interested in the survey topic to participate in the survey. For this survey, respondents had to be 20 years old or older and able to answer the questionnaire in Japanese. The target number of respondents was set at 30,000. To ensure that the collected sample information represented the whole population, we used a quota sampling method based on age (at the time of the survey), gender, and prefectural population ratios obtained from the 2015 National Census [47]. The survey began on 26 February 2021, and ended on 5 March 2021, when it reached the target population set by age, gender, and prefecture, thus, non-response bias is not applicable to the present study. The survey response was on a first-come-first-served basis. The respondents answered all the questions, so there was no missing data. The 53 participants who selected “other” for gender were excluded from the analysis due to the small number of the respondents and difficulty of making reliable estimates. In total, 30,000 participants were included in the analysis. Further details of the survey can be found elsewhere [44].

### 2.2. Surveys

The factors to be included in the questionnaire were determined under the supervision of experts from the Japan Epidemiological Association, the Japanese Society of Infectious Diseases, and the COVID-19 Information Value Improvement and Link project (CIVIL project) after a thorough review of previous research [48,49]. The survey asked the respondents to answer questions about sociodemographic characteristics, health status and literacy, psychological characteristics, their attitude toward immunization and sources of information about COVID-19 and their level of trust. All questions were closed-ended and were asked in single or multiple-answer format, including binary scales, “yes/no” scales, ordinal, nominal, and Likert scales. Unless otherwise noted, the answers are based on the time of survey response. Details of the questionnaire are described elsewhere [44].

### 2.3. Outcome

#### 2.3.1. Preventive Behaviors against COVID-19

Three preventive behaviors were included in this study: 3Cs avoidance, hand hygiene, and respiratory hygiene. Preventive behaviors were measured with the question: “Please choose what you do to prevent COVID-19 infection”. There were ten options and participants could select multiple answers. We have focused on three behaviors: “avoiding the 3Cs”, “hand hygiene” and “respiratory hygiene”. The Japanese government has suggested that people avoid the “3Cs:” “closed spaces”, “crowded places”, and “close-contact settings” [50]. In Japan, the primary health message to prevent the spread of the virus was to “avoid the overlapping of the “3Cs”” [6]. We defined those who answered that they avoided all 3Cs as engaging in preventive behavior. We also included either hand hygiene and respiratory hygiene separately in our analysis based on a review of previous research. Hand hygiene was measured with the option: “hand washing, gargling, and hand disinfection with alcohol”, and respiratory hygiene was measured with the option: “etiquette (handkerchiefs, masks, etc.) when sneezing or coughing”.

#### 2.3.2. Other Variables

Socioeconomic variables include age, gender, education level, prefecture of residence, type of occupation, household income in 2020, household size, marital status, and the degree of impact of the COVID-19 pandemic on one’s life.

With regard to health status and literacy, the respondents were asked about their self-reported health status, the presence of underlying medical conditions, living with elderly family members or family members with underlying medical conditions, their experience with COVID-19 testing, having someone close to them infected with COVID-19, and having refrained from seeking medical examination. As for psychological variables, the respondents provided answers on anxiety about COVID-19, the perceived risk of COVID-19, and their own likelihood of infection within the next six months. These questions were measured based on single items to consider the many variables and to minimize respondent burden. Finally, we asked about the COVID-19 vaccine. Specifically, the respondents were asked about their perception of the benefits, risks and disadvantages of the vaccine against COVID-19. In addition, they were asked whether they should be vaccinated if other people are vaccinated with the COVID-19 vaccine, and whether health care professionals and employees of elder care facilities should be vaccinated with the COVID-19 vaccine. We also investigated their trust in the scientists who developed the COVID-19 vaccine, the approval authority, and the health care workers who conduct the vaccination. The history of routine vaccinations and influenza vaccinations was also examined. Thirty options were provided for sources of information about COVID-19, including medical professionals, literature, television, the Internet, government agencies, family, friends, scientists, companies, social media, and so on. The respondents selected the multiple options they use as information sources on COVID-19. They also rated each of these sources on a 4-point trust scale.

### 2.4. Data Analysis

Odds ratios and 95% confidence intervals were estimated by multivariable logistic regression using a backward-forward stepwise variable selection method, removing those with *p* ≥ 0.1 and adding terms with *p* < 0.05. Model 1 estimates the odds ratios of 3Cs avoidance, Model 2 is used for hand hygiene, and Model 3 is used for respiratory hygiene. To avoid overfitting and multicollinearity, the number of variables was reduced by integrating options for several variables. For example, for information sources, we integrated doctors, nurses, pharmacists, dentists, and veterinarians as “healthcare professionals”; newspapers, TV and radio as “newspaper, TV and radio;” government, local authorities and the Novel Coronavirus Expert Meeting as “public sectors”; and YouTube, LINE, Facebook, Instagram, Twitter and TikTok as “social network services (SNS)”. For trust levels in each information sources, we redefined answers on a scale of 1 (not at all) to 4 (very much) such that they could be treated as continuous variables. The trust level of the integrated groups was calculated as the average of the trust level of each information source included in the group. Regarding the number of information sources used, it was a simple sum, with each source counted as 1 if it was used. For example, if a respondent chose doctors, nurses, and pharmacists as their information sources on COVID-19, the number of information sources used was calculated as three. For the number of social network services used, it was defined in the same way described above. We treated the number of information sources as categorical variables. We have also built other models and treated the variables as continuous variables in order to analyze the trend. Correlations of the variables were set to not exceed 0.8, and the Variable Inflation Factor (VIF) was less than 10. All analyses were performed using STATA/BE version 17.0. Results were considered statistically significant if *p* ≤ 0.05.

## 3. Results

The sociodemographic characteristics of the respondents, stratified by type of preventive measures, are presented in Table 1. The prevalence of preventive behaviors was 35.92% for 3Cs avoidance, 80.43% for hand hygiene, and 67.66% for respiratory hygiene.

### 3.1. Use of and Trust in COVID-19 Information Sources

As shown in Figure 1, the most frequently consulted sources of information were TV (81.63%), followed by internet news sites (53.32%) and newspapers (35.39%). The sources most commonly trusted were doctors (mean 2.73, SD 0.75), followed by nurses (mean 2.60, SD 0.73), and pharmacists (mean 2.47, SD 0.73). The participants mentioned frequent use of TV, news sites and newspapers, but with low levels of trust. Conversely, the respondents reported seeking less information from doctors, nurses, and pharmacists; however, they expressed a higher level of trust in these sources. Comparing local authorities and the national government, local authorities were used and trusted more than the national government (local authorities: usage 28.88%, trust level 2.40, the national government: usage 18.26%, trust level 2.23).

### 3.2. Number of Information Sources Used

Figure 2 indicates the number of information sources used. 22.10% of the respondents chose one, which is the highest percentage, followed by 20.98% who chose two, 17.18% who chose three, and 12.39% who chose four. Figure 3 shows which information source is used for those who use only one source of information. The largest number of respondents used only television (n = 3938). Two hundred sixty-two respondents used only a single social networking service as their information source.

### 3.3. Predictors of Preventive Behaviors

Table 2 indicates the odds of adopting 3Cs avoidance, hand hygiene and respiratory hygiene. For socioeconomic variables, being male was negatively associated with all three of the preventive measures analyzed in this study (3Cs avoidance: OR 0.71, 95% CI 0.66–0.76, hand hygiene: OR 0.65, 95% CI 0.60–0.71, respiratory hygiene: OR 0.61, 95% CI 0.57–0.65). Older age had a positive association with the adoption of the three preventive measures.

For information sources, the usage of healthcare professionals (3Cs avoidance: OR 1.36, 95% CI 1.26–1.47, hand hygiene: OR 1.27, 95% CI 1.13–1.43, respiratory hygiene: OR 1.27, 95% CI 1.16–1.39), TV, radio and newspaper (3Cs avoidance: OR 1.47, 95% CI 1.35–1.61, hand hygiene: OR 2.15, 95% CI 1.96–2.36, respiratory hygiene: OR 1.89, 95% CI 1.74–2.05), the Internet (3Cs avoidance: OR 1.37, 95% CI 1.30–1.46, hand hygiene: OR 1.92, 95% CI 1.79–2.06, respiratory hygiene: OR 1.64, 95% CI 1.55–1.74), social media (3Cs avoidance: OR 1.12, 95% CI 1.04–1.20, hand hygiene: OR 1.44, 95% CI 1.30–1.60, respiratory hygiene: OR 1.48, 95% CI 1.36–1.61), medical information sites (3Cs avoidance: OR 1.31, 95% CI 1.10–1.55, hand hygiene: OR 2.16, 95% CI 1.57–2.97, respiratory hygiene: OR 1.39, 95% CI 1.12–1.72), government agencies (3Cs avoidance: OR 1.42, 95% CI 1.34–1.50, hand hygiene: OR 1.92, 95% CI 1.74–2.11, respiratory hygiene: OR 1.73, 95% CI 1.61–1.85), and family and friends (3Cs avoidance: OR 1.12, 95% CI 1.05–1.20, hand hygiene: OR 1.52, 95% CI 1.36–1.69, respiratory hygiene: OR 1.34, 95% CI 1.24–1.45) were positively associated with the adoption of all the preventive measures included in this study. Trust in social media was associated with the lower odds (3Cs avoidance: OR 0.91, 95% CI 0.86–0.96, hand hygiene: OR 0.73, 95% CI 0.67–0.80, respiratory hygiene: OR 0.81, 95% CI 0.76–0.87), while trust in healthcare professionals was associated with the higher odds for all three measures (3Cs avoidance: OR 1.10, 95% CI 1.05–1.16, hand hygiene: OR 1.12, 95% CI 1.04–1.20, respiratory hygiene: OR 1.13, 95% CI 1.06–1.19).

For other variables, living in urban areas such as Kanto and Kinki, being unmarried without a partner, living with more than two people, and being a homemaker were positively associated with all three preventive measures analyzed in this study (Appendix A). In addition, the stronger the impact of COVID-19 on one’s daily life, the higher the odds of preventive measure adoption (3Cs avoidance: OR 1.60, 95% CI 1.38–1.84, hand hygiene: OR 2.59, 95% CI 2.21–3.02, respiratory hygiene: OR 1.85, 95% CI 1.62–2.12). Regarding health-related factors, higher health literacy had a positive impact on the adoption of public health measures (3Cs avoidance: OR 1.58, 95% CI 1.23–2.04, hand hygiene: OR 2.63, 95% CI 2.05–3.38, respiratory hygiene: OR 1.86, 95% CI 1.48–2.35). Those who have already been infected had significantly lower odds, while those who refrained from seeking medical care due to the COVID-19 situation had higher odds. Concerning psychological variables, as the level of anxiety goes up, the odds of taking preventive measures go up (3Cs avoidance: OR 3.26, 95% CI 2.79–3.82, hand hygiene: OR 2.20, 95% CI 1.84–2.62, respiratory hygiene: OR 1.76, 95% CI 1.51–2.05). COVID 19 vaccine intention was not associated with the adoption of any preventive measures. The odds were higher for those who think that the benefit of the COVID-19 vaccine is large (3Cs avoidance: OR 1.33, 95% CI 1.07–1.67, hand hygiene: OR 1.64, 95% CI 1.28–2.10, respiratory hygiene: OR 1.60, 95% CI 1.29–1.99), and lower for those who think that the vaccine would ease anxiety (3Cs avoidance: OR 0.77, 95% CI 0.61–0.96, hand hygiene: OR 0.72, 95% CI 0.53–0.97, respiratory hygiene: OR 0.50, 95% CI 0.39–0.64). Those who trust public authorities to approve vaccines for COVID-19 were more likely to have lower odds (3Cs avoidance: OR 0.72, 95% CI 0.55–0.96, hand hygiene: OR 0.46, 95% CI 0.32–0.66, respiratory hygiene: OR 0.61, 95% CI 0.45–0.84).

The correlation between the number of information sources used and preventive measures is shown in Table 3 and Appendix A. As the number of information sources used increased, the odds of taking preventive actions increased. For 3Cs avoidance, compared to those who use only one information source, the odds of avoiding 3Cs situations were 1.39 for two sources (95% CI 1.28–1.52), 1.62 for three sources (95% CI 1.48–1.78), 1.92 for four sources (95% CI 1.74–2.11), and 2.70 for five or more sources (95% CI 2.48–2.94). For hand hygiene, compared to those who used only one source of information, those who used multiple information sources had odds of 1.92 for two sources (95% CI 1.76–2.10), 2.70 for three sources (95% CI 2.43–2.99), 3.65 for four sources (95% CI 3.21–4.14), and 5.92 for more than five sources (95% CI 5.27–6.64). For respiratory hygiene, compared to those who used only one source of information, those who used multiple sources of information had odds of 1.68 for two sources (95% CI 1.55–1.81), 2.16 for three sources (95% CI 1.99–2.35), 2.83 for four sources (95% CI 2.57–3.13), and 4.68 for five or more sources (95% CI 4.29–5.11). The linear models showed similar results (3Cs avoidance: OR 1.26, 95% CI 1.24–1.29, hand hygiene: OR 1.54, 95% CI 1.50–1.59, respiratory hygiene: OR 1.44, 95% CI 1.41–1.47).

Table 4 indicates the relationship between the number of social media services used and the adoption of preventive measures. For 3Cs avoidance, the more social media services one used, the higher the odds of avoiding the 3Cs (one source: OR 1.24, 95% CI 1.15–1.35, two sources: OR 1.44, 95% CI 1.23–1.68, three sources: OR 1.76, 95% CI 1.34–2.31, four sources: OR 2.41, 95% CI 1.59–3.65, five sources: OR 2.68, 95% CI 1.53–4.70, six sources: OR 4.55, 95% CI 2.03–10.19). Similarly, for hand hygiene and respiratory hygiene, those who use multiple social media services tend to implement the measures compared to those who utilize a limited number of social media services (hand hygiene: one source: OR 1.76, 95% CI 1.57–1.96, two sources: OR 1.78, 95% CI 1.45–2.20, three sources: OR 1.80, 95% CI 1.26–2.57, four sources: OR 2.19, 95% CI 1.27–3.78, five sources: OR 4.86, 95% CI 1.85–12.79, six sources: OR 3.28, 95% CI 1.03–10.47, respiratory hygiene: one source: OR 1.74, 95% CI 1.59–1.90, two sources: OR 1.96, 95% CI 1.65–2.33, three sources: OR 1.88, 95% CI 1.40–2.53, four sources: OR 3.23, 95% CI 1.99–5.24, five sources: OR 2.32, 95% CI 1.22–4.40, six sources: OR 9.35, 95% CI 2.64–33.13). Similar results were obtained for the linear models (3Cs avoidance: OR 1.22, 95% CI 1.17–1.28, hand hygiene: OR 1.38, 95% CI 1.30–1.46, respiratory hygiene: OR 1.41, 95% CI 1.34–1.48).

## 4. Discussion

The prevalence of preventive behaviors is 35.92% for 3Cs avoidance, 80.43% for hand hygiene and 67.66% for respiratory hygiene. The prevalence of hand hygiene was the highest, which is consistent with prior studies conducted in Japan [51]. The adherence needs to be further improved since hand hygiene, for example, is one of the most cost-effective preventative measures to slow the spread of the virus [52].

Regarding sources of COVID-19-related information, information sources trusted vs. used do not always overlap. Health care professionals were highly trusted information sources, but their usage was low. It might be useful to promote the use of highly reliable information sources to further enhance compliance with COVID-19 preventive measures.

We found that sociodemographic variables were significantly associated with all the studied forms of preventive behaviors. Prior studies have shown that women are more likely to comply with preventive behaviors [6,13,30], which is consistent with our findings. Second, age was also found to be associated with the adoption of all preventive behaviors. According to previous studies, the odds of taking preventive measures increases with age [53]. There was an association between area of residence and the adoption of preventive behaviors. In general, residents in metropolitan areas were more likely to take preventive actions than those in rural areas, which is in line with previous research [36]. These findings suggest that policies against COVID-19 should be targeted at specific gender and age groups and considered on a regional basis.

Consistent with previous studies, levels of anxiety are strongly linked to preventive behaviors [19,54]. In short, the greater the level of anxiety, the higher the odds of taking preventive actions. There is also a strong correlation between the magnitude of COVID-19’s impact on one’s life and preventive behavior. The greater the disruption to one’s life, the higher the odds of taking preventive action, but it is important to note that this is a correlation, not a causal relationship. Among those who have already been infected with the virus, the odds of taking all of the preventive measures were considerably small, which is in line with the previous research [55]. Given that more and more people have been infected with COVID-19, it is crucial to promote preventive measures for those who have already been infected.

Regarding attitudes towards COVID-19 vaccination and preventive behaviors, complex associations were observed. Those who do not believe that vaccination will ease their fears as well as those who do not trust public authorities regarding the approval of vaccines tend to take preventive action. In addition, those who believed that the benefits of vaccines were small had lower odds of taking preventive actions. In order to promote the adoption of preventive measures, it may be valuable to clearly communicate the benefits of vaccination.

Our study revealed that the use of information sources is almost always linked to the adoption of all the preventive behaviors included in this analysis, which is consistent with previous research [56]. Seeking information itself is a positive action, and exposure to diverse sources of information would be meaningful in terms of promoting preventive behaviors.

Prior research has also identified that trust in information sources is an important predictor of preventive behavior [30]. We also found correlations between trust in information sources and prevention behavior. Trust in health care providers is positively associated with all of the preventive actions. Hearing directly from health care professionals might contribute to the adherence to preventive behaviors. According to previous studies, trust in the government was identified to be strongly correlated with prevention behaviors [39]. Consistent with prior research, local governments had a higher level of trust as an information source compared to the national government [6]. This may be due to differences in infection status by region, as differences in the adoption of protective behaviors between urban areas and rural areas are often observed. Thus, local governments should take the initiative in disseminating information depending on the infection situation in their regions.

According to Kusama et al., associations between the usage of information sources and preventive behavior differ according to the type of preventive behaviors [56]. Similarly, our research found that associations between the trust levels of information sources and preventive behaviors differ according to the type of preventive behaviors. It might be meaningful to change channels of information dissemination depending on the type of preventive action required.

Prior research has indicated that trust in social media is associated with vaccination intention [57]. Our study revealed that trust in social media was negatively associated with the adoption of preventive behaviors. On the other hand, the use of social media is positively associated with the adoption of preventive behaviors. Previous studies showed that social media often includes misleading information [32,58]. Misinformation might promote erroneous practices that increase the spread of the virus [59,60,61]. While use of social media could be useful in promoting preventive actions, it is vital to develop literacy. Further investigation is needed to understand how each social media platform influences the practice of preventive measures.

The number of information sources used was identified to be positively correlated with preventive behaviors. In other words, the use of multiple information sources might promote preventive measures. Furthermore, previous research has found a positive correlation between the use of social networking services and social distancing behavior [40]. Our results add to the knowledge that the use of multiple social networking services has a positive relationship with adoption of preventive behaviors.

Based on our research, we have three suggestions to promote COVID-19 preventive behaviors. First, it might be useful to leverage municipalities for COVID-19 communication rather than the national government. As shown in our analysis, people tend to use and trust local governments as a source of information more than the national government. As public health situations differ depending on the region, municipalities might need to take more initiative in COVID-19 communication. Second, it might be beneficial to create opportunities to hear information directly from healthcare professionals. We identified that healthcare professionals are the most trusted source of COVID-19 information in Japan, but they are not fully utilized. Furthermore, trust in healthcare professionals is positively associated with preventive behaviors. They might be one of the key channels to enhance the adherence to COVID-19 preventive measures. Third, improvement in social media literacy is essential. Our research revealed that the use of social media was positively associated with preventive behaviors, while trust in social media was negatively associated with preventive behaviors. It is imperative to improve social media literacy so that people do not trust all social media information at face value. Last but not least, developing messages directed to the demographic characteristics of the different social groups is crucial. The characteristics influence the use and trust of information sources, thus, adopting approaches such as micro-segmented communication might be effective [62].

The present study has several limitations. First, the representativeness of the study participants might have been affected by self-selection bias. While the decision to participate in a survey tends to be influenced by interest in the topic, the fact that respondents were awarded “points” in this survey may have motivated participation even among those who were not interested in the topic of this survey. In addition, it is important to note selection bias and sampling bias, which are common in online surveys [63]. Even though we could not conduct a multitrait–multimethod study to evaluate the reliability and validity, we developed the questionnaire based on the opinions of the experts as well as a thorough review of previous research to ensure content validity. Furthermore, although the demographic distribution of the study population was similar to that of the total population because of the quota sampling method based on age, gender, and prefectural population ratio in the 2015 census, other demographic information such as socioeconomic status and education could not be adjusted. However, none of these socioeconomic variables were significant in the present study in general, so they may not have a significant impact on the results. In addition, Internet-based surveys may result in lower levels of anxiety about certain social problems compared to interviews or other survey methods [64]. Second, as this study was conducted in February–March 2021 when information about the COVID-19 vaccine was limited, participants’ knowledge of and attitudes towards COVID-19 may have changed over time [65]. However, this survey is valuable because it was conducted in the midst of the pandemic under a declared state of emergency when preventive measures were critical to slowing the spread of the virus. In addition, this is a cross-sectional study, thus, it gives limited estimates and implications for the causality. Finally, as this study was only conducted in Japan, the applicability to other cultural environments is limited due to differences in socioeconomic status such as literacy and internet access.

## 5. Conclusions

The present study identified multiple factors related to the adoption of preventive measures. In summary, encouraging access to multiple information sources, utilizing communication channels, and modifying messaging according to target group characteristics are essential to promote COVID-19 preventive measures. While further research is required, these findings could contribute to encouraging the adoption of COVID-19 preventive measures.

## Figures and Tables

**Figure 1 ijerph-19-14511-f001:**
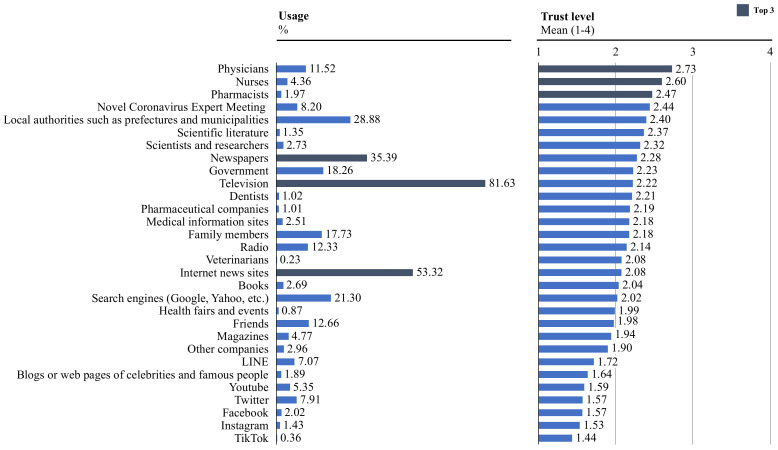
Use and trust levels of COVID-19 information sources (*n* = 30,000).

**Figure 2 ijerph-19-14511-f002:**
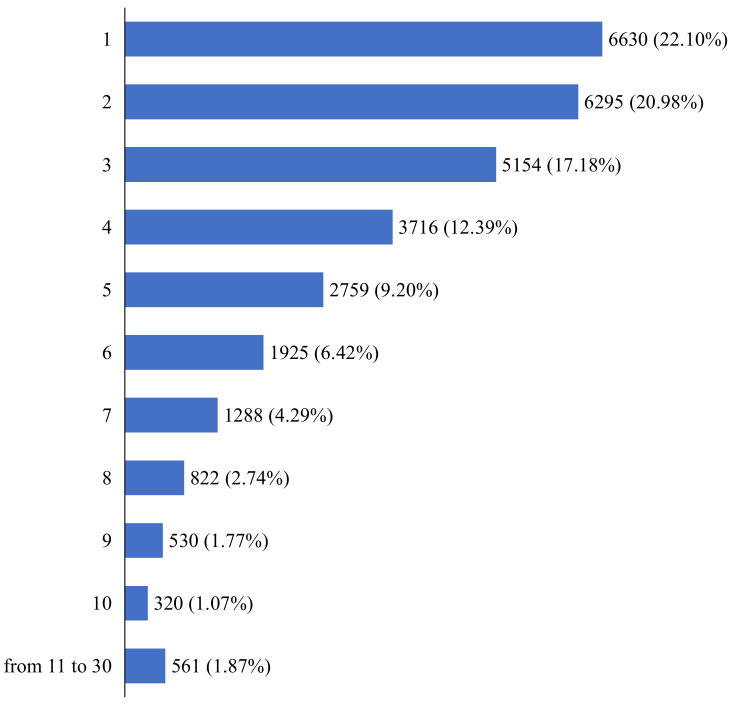
Number of information sources used (*n* = 30,000).

**Figure 3 ijerph-19-14511-f003:**
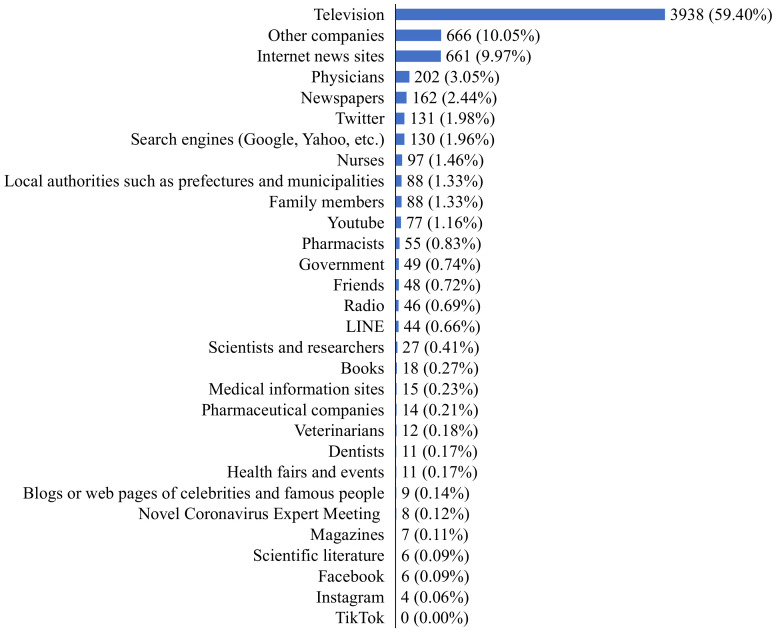
Type of information source used for those who only use one information source (*n* = 6630).

**Table 1 ijerph-19-14511-t001:** Sociodemographic characteristics of the respondents by adoption of preventive measures.

	*n*	3Cs Avoidance	Hand Hygiene	Respiratory Hygiene
Total	30,000	35.92%	80.43%	67.66%
Gender				
Female	15,590	41.89	85.57	74.67
Male	14,410	29.45	74.86	60.08
*p* value		<0.001	<0.001	<0.001
Age				
20s	3491	19.16	64.31	46.66
30s	4460	26.95	73.99	58.74
40s	5312	30.89	77.45	64.85
50s	4483	37.90	82.49	72.41
60s	6168	43.48	86.58	76.52
70s+	6086	47.35	89.24	76.21
*p* value		<0.001	<0.001	<0.001
Highest educational level				
Middle and High school	10,989	35.65	80.38	67.11
Junior college	5831	39.58	84.29	73.28
University	11,866	34.60	79.35	66.19
Graduate school	1314	33.71	73.36	60.58
*p* value		<0.001	<0.001	<0.001

3Cs: “closed spaces”, “crowded places”, and “close-contact settings”.

**Table 2 ijerph-19-14511-t002:** Factors associated with the COVID-19 preventive measures.

	OR (95% CI)		
	3Cs	Hand Hygiene	Respiratory Hygiene
**Age (SA)**			
20s	1.00	1.00	1.00
30s	1.41 (1.25–1.6) ***	1.44 (1.27–1.63) ***	1.55 (1.39–1.73) ***
40s	1.59 (1.41–1.8) ***	1.45 (1.28–1.65) ***	1.84 (1.64–2.05) ***
50s	2.05 (1.81–2.33) ***	1.64 (1.43–1.89) ***	2.35 (2.09–2.66) ***
60s	2.21 (1.94–2.51) ***	1.71 (1.47–1.98) ***	2.51 (2.22–2.84) ***
70s+	2.26 (1.97–2.60) ***	1.78 (1.51–2.11) ***	2.25 (1.96–2.58) ***
Gender (SA)			
Female	1.00	1.00	1.00
Male	0.71 (0.66–0.76) ***	0.65 (0.60–0.71) ***	0.61 (0.57–0.65) ***
Highest educational level (SA)			
Middle and High school	1.00	1.00	1.00
Junior college	1.06 (0.99–1.14)	1.10 (1.00–1.22)	1.14 (1.05–1.24) **
University	1.05 (0.98–1.12)	0.97 (0.90–1.06)	1.02 (0.96–1.09)
Graduate School	1.11 (0.97–1.28)	0.81 (0.69–0.95) *	0.88 (0.76–1.01)
Use of information			
Healthcare professionals	1.36 (1.26–1.47) ***	1.27 (1.13–1.43) ***	1.27 (1.16–1.39) ***
TV, radio and newspaper	1.47 (1.35–1.61) ***	2.15 (1.96–2.36) ***	1.89 (1.74–2.05) ***
Scientific literature	1.36 (1.07–1.72) *	1.48 (1.02–2.16) *	
Books and magazines	1.15 (1.03–1.28) *		1.26 (1.10–1.44) **
The Internet	1.37 (1.30–1.46) ***	1.92 (1.79–2.06) ***	1.64 (1.55–1.74) ***
Medical information sites	1.31 (1.10–1.55) **	2.16 (1.57–2.97) ***	1.39 (1.12–1.72) **
Blogs or web pages of celebrities and famous people		1.34 (0.97–1.86)	
Public sectors	1.42 (1.34–1.50) ***	1.92 (1.74–2.11) ***	1.73 (1.61–1.85) ***
Family and friends	1.12 (1.05–1.20) **	1.52 (1.36–1.69) ***	1.34 (1.24–1.45) ***
Scientists and researchers	1.34 (1.14–1.58) **		1.41 (1.13–1.77) **
Pharmaceutical and other companies		0.82 (0.68–0.98) *	0.83 (0.70–0.98) *
Social media	1.12 (1.04–1.20) **	1.44 (1.30–1.60) ***	1.48 (1.36–1.61) ***
Trust of information			
Healthcare professionals	1.10 (1.05–1.16) **	1.12 (1.04–1.20) **	1.13 (1.06–1.19) ***
Books and magazines		0.88 (0.82–0.95) ***	0.92 (0.86–0.98) **
The Internet	0.96 (0.90–1.01)		1.06 (1.00–1.12)
Medical information sites		1.09 (1.02–1.16) *	1.11 (1.05–1.17) ***
Blogs or web pages of celebrities and famous people		0.90 (0.84–0.98) **	0.91 (0.86–0.97) **
Public sectors		1.15 (1.06–1.24) **	
Family and friends		1.07 (0.99–1.15)	
Scientists and researchers		1.13 (1.04–1.22) **	1.11 (1.05–1.17) ***
Pharmaceutical and other companies	1.08 (1.02–1.15) *	0.91 (0.83–1.01)	
Social media	0.91 (0.86–0.96) **	0.73 (0.67–0.80) ***	0.81 (0.76–0.87) ***

*: *p* < 0.05, **: *p* < 0.01, ***: *p* < 0.001. OR: Odds Ratio, CI: Confidence Intervals. Adjusted for other covariates (see the full regression results in the Appendix A). 3Cs: “closed spaces”, “crowded places”, and “close-contact settings”.

**Table 3 ijerph-19-14511-t003:** Number of information sources used and adoption of preventive measures.

	OR (95% CI)		
	3Cs Avoidance	Hand Hygiene	Respiratory Hygiene
Number of information sources used			
1	1.00	1.00	1.00
2	1.39 (1.28–1.52) ***	1.92 (1.76–2.10) ***	1.68 (1.55–1.81) ***
3	1.62 (1.48–1.78) ***	2.70 (2.43–2.99) ***	2.16 (1.99–2.35) ***
4	1.92 (1.74–2.11) ***	3.65 (3.21–4.14) ***	2.83 (2.57–3.13) ***
5 or more	2.70 (2.48–2.94) ***	5.92 (5.27–6.64) ***	4.68 (4.29–5.11) ***

***: *p* < 0.001. OR: Odds Ratio, CI: Confidence Intervals. Adjusted for other covariates (see the full regression results in the Appendix A). 3Cs: “closed spaces”, “crowded places”, and “close-contact settings”.

**Table 4 ijerph-19-14511-t004:** Number of social network services used and adoption of preventive measures.

	OR (95% CI)		
	3Cs Avoidance	Hand Hygiene	Respiratory Hygiene
Number of social network services used			
0	1.00	1.00	1.00
1	1.24 (1.15–1.35) ***	1.76 (1.57–1.96) ***	1.74 (1.59–1.90) ***
2	1.44 (1.23–1.68) ***	1.78 (1.45–2.20) ***	1.96 (1.65–2.33) ***
3	1.76 (1.34–2.31) ***	1.80 (1.26–2.57) **	1.88 (1.40–2.53) ***
4	2.41 (1.59–3.65) ***	2.19 (1.27–3.78) **	3.23 (1.99–5.24) ***
5	2.68 (1.53–4.70) **	4.86 (1.85–12.79) **	2.32 (1.22–4.40) *
6	4.55 (2.03–10.19) ***	3.28 (1.03–10.47) *	9.35 (2.64–33.13) **

*: *p* < 0.05, **: *p* < 0.01, ***: *p* < 0.001. OR: Odds Ratio, CI: Confidence Intervals. Adjusted for other covariates (see the full regression results in the Appendix A). 3Cs: “closed spaces”, “crowded places”, and “close-contact settings”.

## Data Availability

The datasets generated and/or analyzed during the current study are not publicly available due to ethical considerations but are available from the corresponding author on reasonable request.

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
