# Peer review of "Preventive Behaviors and Information Sources during COVID-19 Pandemic: A Cross-Sectional Study in Japan"

_ijerph, 2022, doi:10.3390/ijerph192114511_

Round 1
Reviewer 1 Report
This paper aims at identifying the factors associated with the adoption of COVID-19 preventive measures, specifically focusing on information sources. This paper used a nationally representative cross-sectional survey of 30,053 Japanese adults to collect data. It then adopted multivariable logistic regression to estimate the factors. Overall, this paper is timely, interesting and well-written. I have several comments as follows.
1. I am delighted to see this study attempts to address a critical issue related to adults’ opinions toward the adoption of COVID-19 preventive measures. However, the current study lacks clear and strong argument and proof for the selection of the factors. In other words, this study proposed many factors that may influence the Japanese adults’ adoption of COVID-19 preventive measures. But, how and why these factors were selected? This study lacks of theoretical explanation and support for these factors.
2. This study adopted self-designed questionnaire to collect data. However, the reliability and validity of the questionnaire can not be guaranteed. The detailed process of questionnaire development needs to be elaborated. Moreover, in order to enhance the reliability of the results, the randomization of the sample selection should also be guaranteed. In another words, how can this paper ensure that the collected sample information can represent the whole population of physicians.
3. Moreover, this study adopted a cross-sectional survey as the data collection method. However, the cross-sectional survey has many biases, such as common method bias and non-response bias. The authors should provide more evidence to control these biases and use statistics to examine the effects of these biases.
Author Response
RESPONSE TO REVIEWERS
We would like to thank the reviewers for your helpful comments. Our responses to the comments are given beneath each comment. The added/revised texts are double quoted in our response for ease of reference, with page and line numbers provided where necessary.
General comments
- This paper aims at identifying the factors associated with the adoption of COVID-19 preventive measures, specifically focusing on information sources. This paper used a nationally representative cross-sectional survey of 30,053 Japanese adults to collect data. It then adopted multivariable logistic regression to estimate the factors. Overall, this paper is timely, interesting and well-written. I have several comments as follows.
We sincerely appreciate your careful reviews and very helpful feedback.
Major comments
- I am delighted to see this study attempts to address a critical issue related to adults’ opinions toward the adoption of COVID-19 preventive measures. However, the current study lacks clear and strong argument and proof for the selection of the factors. In other words, this study proposed many factors that may influence the Japanese adults’ adoption of COVID-19 preventive measures. But, how and why these factors were selected? This study lacks of theoretical explanation and support for these factors.
Thank you for your feedback. The survey questionnaire was developed based on a thorough review of past literature on similar topics and was supervised by the Japan Epidemiological Association and the Japanese Society of Infectious Diseases as well as experts involved in the COVID-19 Information Value Improvement and Link project (CIVIL project). To make this clearer, we have edited the texts as follows:
“The factors to be included in the questionnaire were determined under the supervision of experts from the Japan Epidemiological Association, the Japanese Society of Infectious Diseases, and the COVID-19 Information Value Improvement and Link project (CIVIL project) after a thorough review of previous research 1 2. “ (2.2. Surveys, page 3, line 132)
- This study adopted self-designed questionnaire to collect data. However, the reliability and validity of the questionnaire can not be guaranteed. The detailed process of questionnaire development needs to be elaborated. Moreover, in order to enhance the reliability of the results, the randomization of the sample selection should also be guaranteed. In another words, how can this paper ensure that the collected sample information can represent the whole population of physicians.
Thank you for your important comments. Regarding the questionnaire development process, the details are explained in the response above. Although we could not evaluate the reliability and validity, the questionnaire was developed with reference to the opinions from the experts and previous research. We mentioned this in the limitation section as follows:
“Even though we could not conduct a multitrait–multimethod study to evaluate the reliability and validity, we developed the questionnaire based on the opinions of the experts as well as a thorough review of previous research to ensure content validity”. (4. Discussion, page 12, line 543)
As for the randomization of the sample selection, a quota sampling method based on age, gender, and prefecture population ratios obtained from the 2015 National Census was used. The survey was closed when the number of respondents reached the predetermined target population according to age, gender, and prefecture. To make this clearer, we have revised the texts as follows:
“To ensure that the collected sample information represented the whole population, we used a quota sampling method based on age (at the time of the survey), gender, and prefectural population ratios obtained from the 2015 National Census.” (2.1. Study population, page 3, line 119)
- Moreover, this study adopted a cross-sectional survey as the data collection method. However, the cross-sectional survey has many biases, such as common method bias and non-response bias. The authors should provide more evidence to control these biases and use statistics to examine the effects of these biases.
Thank you for your valuable inputs. As you point out, the representativeness of the study participants might have been affected by biases. As explained in the response to your comment 3, we mentioned this in the limitation section. As for the non-response bias, the target number of study participants was set at approximately 30,000 in this study. Participation was on a first-come-first-served basis and the survey was closed when the number of respondents reached the pre-determined target population according to age, gender, and prefecture, therefore, we believe that the non-response bias is not applicable to the present study. To make this clearer, we have updated the texts as follows:
“The survey began on February 26, 2021, and ended on March 5, 2021, when it reached the target population set by age, gender, and prefecture, thus, non-response bias is not applicable to the present study. The survey response was on a first-come-first-served basis.” (2.1. Study population, page 3, line 122)
- Lin C, Tu P, Beitsch LM. Confidence and Receptivity for COVID-19 Vaccines: A Rapid Systematic Review. Vaccines (Basel) 2020; 9(1).
- Robinson E, Jones A, Lesser I, Daly M. International estimates of intended uptake and refusal of COVID-19 vaccines: A rapid systematic review and meta-analysis of large nationally representative samples. Vaccine 2021; 39(15): 2024-34.

Reviewer 2 Report
The objectives of the study are clearly defined and the methodology is in accordance with the approaches, it is of great value because it objectifies the preventive behaviors in a real scenario of application, and i consider that it has all the merit for its publication because it constitutes a valuable input for the taking of behaviors in public health.
Although the authors acknowledge a selection bias as it is a population that was interested in answering the survey, this probably has an impact on the findings, so a subsequent study is required to try to develop a design that obviates this difficulty.
The sample is very representative, however I doubt the possibility of applicability to other cultural environments such as Latin America where there are still significant shortcomings in the literacy rate. For example, internet access for the general population in my country is still far from complete.
Being a cross-sectional study it only gives us an idea of a moment of reality during the pandemic stage, it would be of great value to be able to make a prospective design at this time of the pandemic or a before and after study to evaluate in the same population the findings on preventive measures. In spite of all this, I find that the manuscript is of great value to be published.
In Colombia, for example, the credibility of some of the media was seriously questioned in the information dissemination strategies https://doi.org/10.26441/RC20.2-2021-A6. On the other hand, the role played by the health authorities in my country coincides with the study with the perception of a greater credibility of the general public to said information. DOI: 10.26633/RPSP.2022.60. Understanding that adherence to handwashing is undoubtedly the most cost-effective strategy, it strikes me that adherence to this measure was not higher during the pandemic (66%), but I think it came close. Much to the estimation of WHO in hospital institutions, I think that this finding should be better treated during the discussion because I consider it to be the most important without ignoring the advantage of bundle stockings.
Author Response
RESPONSE TO REVIEWERS
We would like to thank the reviewers for your helpful comments. Our responses to the comments are given beneath each comment. The added/revised texts are double quoted in our response for ease of reference, with page and line numbers provided where necessary.
General comments
- The objectives of the study are clearly defined and the methodology is in accordance with the approaches, it is of great value because it objectifies the preventive behaviors in a real scenario of application, and i consider that it has all the merit for its publication because it constitutes a valuable input for the taking of behaviors in public health.
We sincerely appreciate your careful reviews and feedback.
Major comments
- Although the authors acknowledge a selection bias as it is a population that was interested in answering the survey, this probably has an impact on the findings, so a subsequent study is required to try to develop a design that obviates this difficulty.
Thank you for your important comments and suggestions for a subsequent study. As you point out, we acknowledge the selection bias. In order to address this, we set the incentives for the respondents to join the panel. Those who responded to questionnaires were provided with ’points’, which could be used to purchase products and services. We believe that the points helped to reduce the selection bias by motivating those who are not interested in the survey topic to participate in the survey. To make it clearer, we have edited the texts as follows:
“Those who respond to surveys are awarded "points" based on the volume of responses they provide, and the points can be used to purchase products and services from partner companies. It is thought to be effective in reducing the effect of the selection bias because it may motivate those who are not interested in the survey topic to participate in the survey.” (2.1. Study population, page 2, line 96)
- The sample is very representative, however I doubt the possibility of applicability to other cultural environments such as Latin America where there are still significant shortcomings in the literacy rate. For example, internet access for the general population in my country is still far from complete.
We appreciate for your inputs. We totally agree that the applicability of this study to other cultural environments is limited. The present study is conducted only in Japan, therefore, it might not be applicable to other cultures where the internet access is far from complete. To make this clearer, we have revised the texts as follows:
“Finally, as this study is conducted only in Japan, the applicability to other cultural environments is limited due to the difference in the socioeconomic status such as literacy and internet access.” (4. Discussion, page 12, line 559)
- Being a cross-sectional study it only gives us an idea of a moment of reality during the pandemic stage, it would be of great value to be able to make a prospective design at this time of the pandemic or a before and after study to evaluate in the same population the findings on preventive measures. In spite of all this, I find that the manuscript is of great value to be published. In Colombia, for example, the credibility of some of the media was seriously questioned in the information dissemination strategies https://doi.org/10.26441/RC20.2-2021-A6. On the other hand, the role played by the health authorities in my country coincides with the study with the perception of a greater credibility of the general public to said information. DOI: 10.26633/RPSP.2022.60. Understanding that adherence to handwashing is undoubtedly the most cost-effective strategy, it strikes me that adherence to this measure was not higher during the pandemic (66%), but I think it came close. Much to the estimation of WHO in hospital institutions, I think that this finding should be better treated during the discussion because I consider it to be the most important without ignoring the advantage of bundle stockings.
Thank you for introducing the situation in Columbia. We agree that hand washing is one of the most cost-effective strategies. In this study, the adherence to hand hygiene was 80%, but it can be further improved. We have mentioned this in the discussion section as follows:
“The adherence needs to be further improved since preventive measures such as hand hygiene is one of the most cost-effective measures to slow the spread of the virus 1 .” (4. Discussion, page 10, line 446)
- Juneau CE, Pueyo T, Bell M, Gee G, Collazzo P, Potvin L. Lessons from past pandemics: a systematic review of evidence-based, cost-effective interventions to suppress COVID-19. Syst Rev 2022; 11(1): 90.

Reviewer 3 Report
The paper addresses a relevant issue. The literature review, however, is insufficient, especially at the beginning of the text.
The authors refer to few studies on the "number" of information sources, although several are about the "type". They also regard the "trust" in these sources of information about Covid-19.
The literature review could problematize "trust" (a predictor of preventive behaviours in the health area), making the investigation more sound. As the authors conclude, trust in information sources is fundamental for promoting health literacy, namely the adoption of prevention behaviours. This issue of "trust" is related to another aspect identified, the need to improve media literacy concerning the ability to identify misinformation conveyed by social media.
Nevertheless, the statistical analysis developed responds to the objectives proposed by the researchers, like identifying the use of multiple sources of information (versus only one). It also allows for studying socioeconomic variables associated with adopting preventive measures.
However, these results are exploratory and can be further developed, especially if there is a better problematization of concepts such as "anxiety" and how it relates, for example, to media consumption and media literacy. It is a multidimensional concept (as, incidentally, "trust"). Is anxiety due only to personal characteristics? Does it relate to the type of source of information used? Is it a result of overexposure to the information? Or does the crossing of information sources help to reduce anxiety? What are the various elements that contribute to "trust" in a given source of information? The ethos (will it be the case of doctors, for example)? What about newspapers versus television? Or local versus national government? In the section "discussion", the authors refer to previous studies to contextualize the results obtained in this investigation. But if this conceptualization were more in-depth initially, it would have contributed to a methodological design that allowed more relevant results. In other words, the discussion on how the data contributes to the existing knowledge in this area, in some way, contributes to lessening the criticism we make regarding the literature review. Still, it doesn't completely rule it out.
The conclusion regarding the need to develop communication directed to the demographic characteristics of the different social groups is adequate. It also can be further developed concerning how these characteristics influence the media diet and the trust in the different types of information. A well-known example is the use of micro-segmented communication, based on personality traits, in areas such as political or marketing communication through social media.
Author Response
RESPONSE TO REVIEWERS
We would like to thank the reviewers for your helpful comments. Our responses to the comments are given beneath each comment. The added/revised texts are double quoted in our response for ease of reference, with page and line numbers provided where necessary.
Major comments
- The paper addresses a relevant issue. The literature review, however, is insufficient, especially at the beginning of the text.
The authors refer to few studies on the "number" of information sources, although several are about the "type". They also regard the "trust" in these sources of information about Covid-19.
The literature review could problematize "trust" (a predictor of preventive behaviours in the health area), making the investigation more sound. As the authors conclude, trust in information sources is fundamental for promoting health literacy, namely the adoption of prevention behaviours. This issue of "trust" is related to another aspect identified, the need to improve media literacy concerning the ability to identify misinformation conveyed by social media.
We sincerely appreciate your careful reviews and very helpful feedback. We totally agree that the trust in information sources is fundamental for promoting the adoption of preventive behaviours. To make this clearer, we have added following texts:
“Trust is one of the predictors of preventive behaviours in health area, thus, understanding the dynamics of how levels of trust in information sources influence preventive behaviors is important for COVID-19 health messaging strategy 1.“ (1. Introduction, page 2, line 59)
- Nevertheless, the statistical analysis developed responds to the objectives proposed by the researchers, like identifying the use of multiple sources of information (versus only one). It also allows for studying socioeconomic variables associated with adopting preventive measures.
However, these results are exploratory and can be further developed, especially if there is a better problematization of concepts such as "anxiety" and how it relates, for example, to media consumption and media literacy. It is a multidimensional concept (as, incidentally, "trust"). Is anxiety due only to personal characteristics? Does it relate to the type of source of information used? Is it a result of overexposure to the information? Or does the crossing of information sources help to reduce anxiety? What are the various elements that contribute to "trust" in a given source of information? The ethos (will it be the case of doctors, for example)? What about newspapers versus television? Or local versus national government? In the section "discussion", the authors refer to previous studies to contextualize the results obtained in this investigation. But if this conceptualization were more in-depth initially, it would have contributed to a methodological design that allowed more relevant results. In other words, the discussion on how the data contributes to the existing knowledge in this area, in some way, contributes to lessening the criticism we make regarding the literature review. Still, it doesn't completely rule it out.
Thank you very much for your important comments. As you point out, anxiety is one of the crucial concepts to understand the mechanism regarding the adoption of individual preventive behaviour. However, the scope of the present study is to investigate the relationship between the adoption of preventive measures and information sources, therefore, we have limited consideration regarding anxiety. Actually, in another study, we have examined the impact of anxiety in details using the same dataset (DOI: 10.1016/j.ssmph.2022.101105). Following is the findings from the research:
・Those with a higher perceived risk of the COVID-19 vaccine had higher odds of risk perception for both infection and severe illness.
・Those with higher odds of risk perception of being infected were more likely to report obtaining their information from healthcare workers whereas those with lower odds were more likely to report obtaining their information from the Internet or the government.
・Those with lower odds of risk perception of being severely ill were more likely to report obtaining their information from the Internet.
・The higher the trust level in the government as a COVID-19 information source, the lower the odds of both risk perception of being infected and becoming severely ill.
・The higher the trust levels in social networking services as a COVID-19 information source, the higher the odds of risk perception of becoming severely ill.
To address this clearer, we have revised the texts as follows:
“In particular, the degree of anxiety is strongly correlated with the adoption of preventive measures as well as with the source of COVID-19 information and its trust level 2.” (1.Introduction, page2, line 56)
- The conclusion regarding the need to develop communication directed to the demographic characteristics of the different social groups is adequate. It also can be further developed concerning how these characteristics influence the media diet and the trust in the different types of information. A well-known example is the use of micro-segmented communication, based on personality traits, in areas such as political or marketing communication through social media.
Thank you for your insights. Based on your feedback, we have further developed the implication regarding the heath messaging as follows:
“Last but not least, developing messages directed to the demographic characteristics of the different social group is crucial. The characteristics influence the use and trust of information sources, thus, adopting approaches such as micro-segmented communication might be effective 3.“(4. Discussion, page 12, line 533)
- Figueiras MJ, Ghorayeb J, Coutinho MVC, Maroco J, Thomas J. Levels of Trust in Information Sources as a Predictor of Protective Health Behaviors During COVID-19 Pandemic: A UAE Cross-Sectional Study. Front Psychol 2021; 12: 633550.
- Adachi M, Murakami M, Yoneoka D, et al. Factors associated with the risk perception of COVID-19 infection and severe illness: A cross-sectional study in Japan. SSM Popul Health 2022; 18: 101105.
- Bailey A, Harris MA, Bogle D, et al. Coping With COVID-19: Health Risk Communication and Vulnerable Groups. Disaster Med Public Health Prep 2021: 1-6.

Reviewer 4 Report
The topic is unique and sheds light on a different perspective of preventive behaviors and information sources during the Covid-19 pandemic. The article is well grounded, based on the theory of planned behaviour which posits a direct impact of individual’s behavioral strategies to cope up with the Covid-19 pandemic.
The methodology part is sound and carried out using quantitative survey filled by 30,053 adults in Japan. The survey asked about socioeconomic, health-related, psychologic characteristics, attitudes toward immunization, and the use of information sources regarding COVID-19.
The results are detailed comprehensively, indicating that the more information sources are available to people, the more cautious they will be regarding taking preventive measures for Covid-19.
The References used in the research are relevant and also carefully selected from the latest literature in the field.
However, the plagiarism is almost 21%. The manuscript can be accepted for publication if the plagiarism is brought down to 15%.
Author Response
RESPONSE TO REVIEWERS
We would like to thank the reviewers for your helpful comments. Our responses to the comments are given beneath each comment. The added/revised texts are double quoted in our response for ease of reference, with page and line numbers provided where necessary.
General comments
- The topic is unique and sheds light on a different perspective of preventive behaviors and information sources during the Covid-19 pandemic. The article is well grounded, based on the theory of planned behaviour which posits a direct impact of individual’s behavioral strategies to cope up with the Covid-19 pandemic.
The methodology part is sound and carried out using quantitative survey filled by 30,053 adults in Japan. The survey asked about socioeconomic, health-related, psychologic characteristics, attitudes toward immunization, and the use of information sources regarding COVID-19
The results are detailed comprehensively, indicating that the more information sources are available to people, the more cautious they will be regarding taking preventive measures for Covid-19.
The References used in the research are relevant and also carefully selected from the latest literature in the field.
We sincerely appreciate your careful reviews and very helpful feedback.
Major comments
- However, the plagiarism is almost 21%. The manuscript can be accepted for publication if the plagiarism is brought down to 15%.
Thank you for pointing out this important issue. We checked the texts ad revised thoroughly.

Round 2
Reviewer 3 Report
The text has improved with the changes introduced.
Nevertheless, the literature review would benefit from a deeper conceptualization of "trust".
Author Response
RESPONSE TO REVIEWERS
We would like to thank the reviewers for your helpful comments. Our responses to the comments are given beneath each comment. The added/revised texts are double quoted in our response for ease of reference, with page and line numbers provided where necessary.
Response to Reviewer
General comments
- The text has improved with the changes introduced.
We sincerely appreciate your careful reviews and very helpful feedback.
Major comments
- Nevertheless, the literature review would benefit from a deeper conceptualization of "trust".
Thank you for your important feedback. As you point out, “trust” is a key predictor of preventive behaviours in the health area, and it plays a vital role in promoting the adoption of preventive measures. To elaborate more on this critical topic, we have reviewed the previous research again and revised the texts as follows:
“Trust is one of the key predictors of preventive behaviours in health area 1. According to Nakayachi and Cvetkovich, trust is composed of competence, fairness, and salient value similarity 2. Trust influences the perception of risk and benefit, accordingly, has a powerful impact on the decision making 3 4. Thus, trust mediates the relationship between information use and health behaviors 5 6 7. Previous study found a positive association between trust in formal information sources and preventative behaviours 8. Previous research in Australia also identified that trust in health care professionals and scientists was linked to greater engagement in protective behaviors 9. Hence, understanding the dynamics of how trust in information sources influence preventive behaviors is essential for developing COVID-19 health messaging strategy. “
- Figueiras MJ, Ghorayeb J, Coutinho MVC, Maroco J, Thomas J. Levels of Trust in Information Sources as a Predictor of Protective Health Behaviors During COVID-19 Pandemic: A UAE Cross-Sectional Study. Front Psychol 2021; 12: 633550.
- Nakayachi K, Cvetkovich G. Public trust in government concerning tobacco control in Japan. Risk Anal 2010; 30(1): 143-52.
- Siegrist M, Cvetkovich G, Roth C. Salient value similarity, social trust, and risk/benefit perception. Risk Anal 2000; 20(3): 353-62.
- Visschers VH, Siegrist M. How a nuclear power plant accident influences acceptance of nuclear power: results of a longitudinal study before and after the Fukushima disaster. Risk Anal 2013; 33(2): 333-47.
- Liao Q, Cowling B, Lam WT, Ng MW, Fielding R. Situational awareness and health protective responses to pandemic influenza A (H1N1) in Hong Kong: a cross-sectional study. PLoS One 2010; 5(10): e13350.
- Bults M, Beaujean DJ, de Zwart O, et al. Perceived risk, anxiety, and behavioural responses of the general public during the early phase of the Influenza A (H1N1) pandemic in the Netherlands: results of three consecutive online surveys. BMC Public Health 2011; 11: 2.
- Blair RA, Morse BS, Tsai LL. Public health and public trust: Survey evidence from the Ebola Virus Disease epidemic in Liberia. Soc Sci Med 2017; 172: 89-97.
- Bradley DT, Mansouri MA, Kee F, Garcia LMT. A systems approach to preventing and responding to COVID-19. EClinicalMedicine 2020; 21: 100325.
- Faasse K, Newby J. Public Perceptions of COVID-19 in Australia: Perceived Risk, Knowledge, Health-Protective Behaviors, and Vaccine Intentions. Front Psychol 2020; 11: 551004.